# Electric Double Layer in Water-Organic Mixed Solvents: Titania in 50% Ethylene Glycol

**DOI:** 10.3390/molecules27072162

**Published:** 2022-03-27

**Authors:** Marek Kosmulski, Edward Mączka

**Affiliations:** Laboratory of Electrochemistry, Lublin University of Technology, 20618 Lublin, Poland; e.maczka@pollub.pl

**Keywords:** electrophoresis, stability of dispersion, nonaqueous solvents, surfactants

## Abstract

Ethylene glycol (EG) and its mixtures with water are popular components of nanofluids used as heat transfer fluids. The stability of nanofluids against coagulation is correlated with their zeta potential. The electrophoretic mobility of titania nanoparticles in 50-50 *w*/*w* EG was studied as a function of the concentration of various solutes. HCl, NaOH, SDS and CTMABr at concentrations up to 0.01 M are strong electrolytes in 50% EG, that is, the conductance of their solutions is proportional to the concentration. HCl, NaOH and CTMABr were very efficient in inducing a high zeta potential for titania in 50% EG. NaOH induced a negative zeta potential in excess of 70 mV, and HCl and CTMABr induced a positive zeta potential in excess of 70 mV at concentrations below 10^−4^ M. Apparently, HCl, NaOH and CTMABr are also more efficient than SDS in terms of nanofluid stabilization against coagulation. An overdose of base (>1 mM) results in depression of the negative zeta potential. This result may be due to the specific adsorption of sodium on titania from 50% EG.

## 1. Introduction

Most studies of the electric double layer have been conducted in water, and studies in mixed water + organic solvents are less numerous. In principle, the models used in aqueous systems also apply to mixed water + organic solvents, but in the experimental procedures and calculations, the specificity of mixed solvents must be taken into account.

The pH-dependent surface charge has been extensively studied. Many materials carry a positive surface charge at a low pH and a negative surface charge at a high pH. Similar pH-dependent surface charging occurs in mixed water + organic solvents, but the measurement and even the definition of pH in such mixed solvents is a complicated problem. We discussed this problem in a series of papers devoted to pH-dependent surface charging in mixed solvents [1,2,3,4].

Electrophoresis is especially useful in studies of the electric double layer in mixed solvents. The results of electrophoretic measurements are usually presented in terms of the zeta potential. The zeta potential is directly related to the stability of dispersions against coagulation and sedimentation. It should be emphasized that there are also factors other than zeta potential affecting the stability. The technical problems encountered in electrokinetic studies of dispersions in mixed solvents, especially in terms of the relationship between zeta potential and stability, were discussed in a recent paper [5]. In our opinion, the following problems have not received sufficient attention in previous studies [6,7,8,9,10,11,12,13,14,15,16,17,18]:The calculation of zeta potential from electrophoretic mobility is not straightforward.The relationship between zeta potential and stability is complicated.More attention should be paid to proper selection of the surfactant and of the surfactant dose.

We address these problems in our study. We focused on mixed solvents based on a mixture of ethylene glycol (EG) and water in this study. Such mixed solvents have been studied as components of nanofluids. We agree with the previous authors who studied EG-based nanofluids that the electrokinetic studies of such nanofluids can be useful in predicting the stability of such nanofluids against coagulation and sedimentation. However, we disagree with the previous authors in several points. First of all, the determination of zeta potential from experimental data requires a special approach. We explain this problem in detail and show the solutions. In particular, we think that most apparent zeta potentials reported for EG and for EG–water mixtures in the literature are severely underestimated (by a factor up to 1.5). We also think that proper determination of the optimum surfactant dose can substantially improve the stability of nanofluids. In many studies of surfactant-stabilized nanofluids, the surfactant dose was arbitrary, and no systematic study of the effect of surfactant dose was undertaken. Such research is needed in view of cost efficiency, that is, excessive use of surfactants implies additional costs, but also for environmental reasons, that is, excessive use of surfactants means higher pollution. Moreover, excessive use of surfactants may lead to lower zeta potentials (in absolute value) and lower stability. We also think that pH-dependent surface charging is underestimated in the studies of potential heat transfer fluids. High zeta potentials and high stabilities can be achieved by the addition of acids or bases with or without the simultaneous addition of surfactants.

A few examples of recent electrokinetic studies in EG–water mixtures are summarized in Table 1. These systems were studied in the context of the stability of heat transfer fluids against coagulation and sedimentation. EG and EG–water mixtures are frequently used in heat transfer fluids, and solid particles are added to improve their heat conductance.

Different solvent compositions were studied, but 50-50 EG–water [9,10,11,12,13] seems to be the most popular one. Therefore, we chose 50-50 EG–water mixture as the solvent in our study. Multiple electrokinetic studies have been carried out in anhydrous EG, but those are beyond the scope of the present paper.

Many electrokinetic studies presented in Table 1 were devoted to dispersions of solid particles in “pure” mixed solvents [7,8,11,13,14,17,18]. On top of the studies performed only on “pure” mixed solvents, several papers presenting the results obtained for surfactant-modified dispersions also show results obtained for “pure” mixed solvents as a reference. In our opinion electrokinetic measurements in “pure” mixed solvents are of limited significance. The surface charge in such systems is due to the presence of impurities of unknown nature and at unknown concentrations in “pure” solvents, thus the results will vary from one lot of solvent to another. This problem is discussed in detail elsewhere [19].

Many electrokinetic studies were carried out in the presence of ionic surfactants [6,9,12,15,16]. Cationic surfactants induce positive zeta potentials of solid particles in 50% EG, and anionic surfactants induce negative zeta potentials. SDS [9,15,16] and CTAB [6,9,12,15,16] are among the most popular ionic surfactants used in the studies of potential heat transfer fluids, and we also used these surfactants in our study. 

The zeta potential of particles dispersed in surfactant solutions depends on the equilibrium concentration of the surfactant in the solution. The problem is that the electrokinetic studies are seldom accompanied by measurements of surfactant adsorption. Thus, the total amount of surfactant in the system is known, but its equilibrium concentration in the solution is not. Two different approaches to this problem can be found in the literature. In most studies presented in Table 1, the surfactant concentration is related to the amount of solid [6,9,12,15,16]. In a few other studies, the surfactant concentration is related to the amount of solution. We use the later approach in our study. Actually, neither approach makes it possible to properly normalize the results unless the distribution of the surfactant between the solution and the surface is known. The problem of normalization is not unique to heat transfer fluids, and it affects all kinds of electrokinetic studies performed in the presence of surfactants.

The solid load (similar dispersions studied at various solid-to-liquid ratios) [12,14,18] and aging (similar dispersions studied at various times after their preparation) [9,12,14,15,16,17,18] were among the most popular variables in the electrokinetic studies of potential heat transfer fluids. Usually, the aging times were on the order of a few weeks, but even 1-year aged dispersions were studied [17]. Most studies showed that the effects of the solid load and aging on the zeta potential were rather insignificant. Therefore, we focus on the effect of the surfactant concentration in the solution in this study. Table 1 presents only the most common variables studied. For example, in [8], zeta potentials are reported in dispersions ultrasonified for different times. We measured the electrophoretic mobility at various (total) surfactant concentrations. We also studied pH-dependent surface charging in 50% EG. As the pH measurement procedure in 50% EG is not readily available, we expressed the acidity of solutions as the molar concentrations of acids and bases. 

## 2. Results and Discussion

### 2.1. Conductance

In order to estimate the ionic strength of the solutions, we measured the conductance of dispersions. The results are presented in Figure 1. Only the conductances obtained at high electrolyte concentrations are presented and analyzed. At high electrolyte concentrations, the effect of solid particles on the conductance is negligible, and the conductance of dispersion is practically equal to the conductance of the solution. 

The conductance is proportional to the concentrations of CTMABr, SDS, HCl and NaOH up to 0.01 M (correlation coefficients > 0.999). Apparently, these electrolytes are fully dissociated in 50% EG in this concentration range. At CTMABr concentrations >0.01 M, a substantial deviation from linearity was observed. These deviations indicate ion pairing or other forms of association. On top of the linearity shown in Figure 1, Walden’s rule is another argument for the full dissociation of CTMABr, SDS, HCl and NaOH up to 0.01 M in 50% EG. The slopes of the straight lines connecting the points representing CTMABr, SDS, HCl and NaOH are lower than the corresponding limiting molar conductances of these electrolytes in water by a factor of 3.4, 2.58, 2.64 and 3.57, respectively, The limiting conductances were calculated from the data from [20]. All these numbers are lower than 3.76, which is the viscosity ratio between 50% EG and water. In other words, the conductances of the solutions of these electrolytes in 50% EG are even higher than predicted from Walden’s rule. 

### 2.2. Debye Length and the Henry Coefficient

As we know the ionic strength, we can also estimate the Debye length and κa. We take 20 nm as the radius of the primary particles. We need κa to calculate the zeta potential from the electrophoretic mobility. A few κa values for typical concentrations c used in this study are summarized in Table 2. The κa values presented in Table 2 are independent of the nature of the electrolyte. 

Table 2 also presents the Henry coefficient in the following equation:
*ζ* = 1.5 *μη*/*ε*/*f*,(1)
where *ζ* is the zeta potential, *μ* is the electrophoretic mobility, *η* is the viscosity, *ε* is the dielectric constant and *f* is a coefficient, which depends on *κa*. With low *κa*, *f* approaches 1 and Equation (1) becomes a Huckel equation. With high *κa*, *f* approaches 1.5 and Equation (1) becomes a Smoluchowski equation. Equation (1) is only valid for spherical particles at low *ζ* (<40 mV in absolute value). With higher *ζ*, Equation (1) is only a rough estimation, but it is still a better estimation than the Huckel or Smoluchowski equations. Equation (1) underrates the zeta potential when it is high in absolute value. There are more elaborate equations to convert mobility into zeta potential [21], but they are only valid for spherical particles and require data which are not available for 50% EG. Therefore, we use Equation (1) in this paper. Table 2 indicates that exact knowledge of *κa* is not essential for the determination of *f*. A difference in *κa* of a few percent has a rather insignificant effect on *f*.

### 2.3. Zeta Potential 

The zeta potential of titania in “pure” 50% EG was 9.3 mV, with a standard deviation of 0.7 mV. The decimal digit in “9.3 mV” is not significant, and the calculated zeta potentials should be rounded to full millivolts. We do not pay much attention to the zeta potential in “pure” 50% EG. First of all, our titania contains traces of HCl, according to the manufacturer (indeed the aqueous dispersion is acidic), and the presence of acid in the dispersion may be responsible for the positive *ζ*. Moreover, the value of *ζ* potential in “pure” 50% EG may also be affected by impurities of unknown nature present at unknown concentrations in reagent-grade EG, and dispersions prepared with another lot of EG may show different *ζ* potentials, as discussed in the Introduction. We use the zeta potential in “pure” 50% EG as a reference in measurements of *ζ* potentials in the presence of solutes.

The pH-dependent surface charging of titania in 50% EG is shown in Figure 2. According to expectations, HCl induces a positive *ζ* potential and NaOH induces a negative *ζ* potential. The HCl concentration of 0.00005 M induces a *ζ* potential as high as +80 mV, and the further addition of acid has a rather insignificant effect on the *ζ* potential. The NaOH concentration of 0.00005 M induces a negative *ζ* potential as high as −80 mV. The further addition of base enhances the *ζ* potential, but with 0.001 M base the trend reverses, that is, the further addition of base depresses the *ζ* potential. The addition of base leads to an increase in the ionic strength. It is well known (cf. [3] and references therein) that the high ionic strength depresses the absolute value of ζ potential. Moreover, monovalent cations are specifically adsorbed on titania from water-alcohol mixtures (cf. [3] and references therein), and it can very well be that similar specific adsorption occurs in EG–water mixtures. The specific adsorption of cations leads to a more positive (less negative) electrokinetic charge. Thus, we have two opposite trends in the addition of base: increases in the pH lead to more negative ζ potential, but increases in the ionic strength and specific adsorption of cations lead to less negative ζ potential. Apparently, the pH effect prevails at low base concentrations, but the ionic strength effect and/or the specific adsorption of cations prevail at high base concentrations. Figure 2 indicates that the addition of acids or bases is a very efficient way to produce highly charged particles in 50% EG. Apparently, the possibility of producing highly charged particles by the addition of acids or bases was underestimated by the previous researchers, who preferred to use surfactants (Table 1). Charge regulation by the addition of acids or bases is also cost-efficient and environment-friendly as compared with charge regulation by the addition of surfactants. For example (Sigma-Aldrich online catalogue, accessed 21 March 2022), 1 mole of NaOH (reagent grade, 1 kg jar) costs 1.6 euro, while 1 mole of SDS (reagent grade, 1 kg jar) costs 28 euro and 1 mole of CTMABr (reagent grade, 1 kg jar) costs 106 euro. Many papers have been published on the environmental risks related to surfactants ([22], and references therein).

The effect of ionic surfactants on the *ζ* potential of titania in 50% EG is shown in Figure 3. According to expectations, a cationic surfactant induces a positive *ζ* potential and an anionic surfactant induces a negative *ζ* potential. The CTMABr concentration of 0.00002 M induces a *ζ* potential as high as +94 mV, and further addition of CTMABr has a rather insignificant effect on the *ζ* potential. In contrast, the negative *ζ* potentials gradually increased with the addition of SDS over the entire studied range, and the surfactant concentration required for a *ζ* potential of 100 mV in absolute value is higher in SDS than in CTMABr by several orders of magnitude. The substantial difference between SDS (HLB of 40) and CTMABr (HLB 10) may be due to the difference in their hydrophylic-lipophylic characters.

The electrokinetic behavior of titania in the presence of ionic surfactants in aqueous systems is pH-dependent, and we suppose that this is also the case in 50% EG. Figure 2 provides a clear evidence that the pH matters. We do not know how to measure pH in 50% EG, but we performed electrokinetic measurements in systems containing both SDS and base. We know that the original titania contains HCl, and thus that the dispersion containing titania and surfactant (no acids or bases added) is acidic. Figure 4 presents the zeta potential in the system, in which the acid contained in the titania was neutralized with NaOH. The analogous results obtained without the addition of base are given as a reference. The addition of base enhanced the negative zeta potentials obtained in the presence of SDS over the surfactant concentration range 10^−4^–10^−3^ M, but at [SDS] > 4 × 10^−3^ M, the difference between the zeta potentials obtained in the presence and absence of base is insignificant. Interestingly enough, the zeta potentials obtained in the presence of about 10^−4^ SDS and 0.00004 M NaOH are less negative than the zeta potentials obtained in the presence of 0.00004 M NaOH alone (no surfactant). In other words, the addition of SDS depressed the negative zeta potential in these systems. In view of a correlation between the absolute value of zeta potential and dispersion stability, we think that CTMABr is a better agent to stabilize titania dispersions in 50% EG, with or without the addition of base. 

## 3. Materials and Methods

Aeroxide P25 was a titania from Degussa-Evonik, chiefly anatase. It contains <0.3% of alumina, <0.2% of silica and trace amounts of HCl (<0.3%). The specific surface area is 50 m^2^/g, the particle size is 20 nm and the isoelectric point is at pH 6.5. P25 is hygroscopic and contains variable amounts of water unless dried and stored in an air-tight bottle.

EG was analytical grade from Standard, Lublin. SDS and CTMABr were from Sigma-Aldrich. The other chemicals were from POCh, Lublin, Poland. All the chemicals were used as received.

50% EG by mass was prepared gravimetrically. 

The solutions of surfactants 5 g/L were prepared gravimetrically from 50% EG and dry surfactants. More diluted solutions were prepared on *v*/*v* basis by dilution of the above concentrated solutions with 50% EG. We also prepared 0.5 g/L solutions by dilution of 5 g/L solutions with 50% EG, and then prepared very diluted solutions on a vol/vol basis by dilution of the 0.5 g/L solutions with 50% EG. 

The 0.1 M solutions of acid and base in 50% EG were prepared gravimetrically from EG, 1 M aqueous acid or base and water. To this end, we mixed 5 g of EG, 4 g of water and 1 g of 1 M aqueous acid or base. More diluted solutions were prepared on a vol/vol basis by dilution of the above concentrated solutions with 50% EG. We also prepared 0.01 M solutions by dilution of 0.1 M solutions with 50% EG, and then prepared very diluted solutions on a *v*/*v* basis by dilution of the 0.01 M solutions with 50% EG. 

We did not attempt to measure the pH in our dispersions. The pH measurements in nonaqueous systems and even the definition of pH are difficult, as explained in the Introduction. Therefore, we only report the analytical concentrations of acid and base.

The dispersions had a solid load of 1:10,000 by mass. The electrophoretic mobility and conductance were measured at 25 °C by means of a Malvern Zetasizer. The measurement was repeated 3 times for each dispersion. 

The electrophoretic mobility was converted into zeta potential by means of Equation (1). We do realize that the Henry equation is not suitable for calculation of high zeta potentials. Most likely, the zeta potentials reported in the figures are underestimated in absolute value.

We needed the viscosity and dielectric constant of 50% EG to calculate the zeta potential (cf. Equation (1)). The viscosities of aqueous EG have been extensively studied [23,24,25]. In contrast, the data on the dielectric constant of aqueous EG are limited, and even relatively recent publications [26] only cite a 90-year-old paper by Akerlof [27] as the only source. Critical analysis of the available data led us to the following values (50% EG, 25 °C): a viscosity of 3.35 cP and a dielectric constant of 51. The latter value was based on the original data for EG–water mixtures [27] obtained at 20 °C and the temperature coefficients taken from [20,26].

## 4. Conclusions

Although the present study was carried out for one solvent (50% EG), one type of solid particle (P25 titania) and two surfactants (SDS and CTMABr), the results presented above can probably be generalized:for different EG–water ratios, e.g., 40 or 60% EG;for mixtures of other glycols, e.g., of propanediols with water;for other solids showing pH-dependent surface charging, e.g., alumina and iron (hydr)oxides;for other ionic surfactants.

We speculate that in such systems, zeta potentials on the order of plus or minus 100 mV can be induced by the addition of acid, base or ionic surfactants. Relatively low acid, base or surfactant concentrations induce a zeta potential on the order of plus or minus 100 mV, and further increases in their concentrations has a rather insignificant effect on the zeta potential. The results of this study can be useful in the design of the recipes of nanofluids, especially for heat transfer fluids.

## Figures and Tables

**Figure 1 molecules-27-02162-f001:**
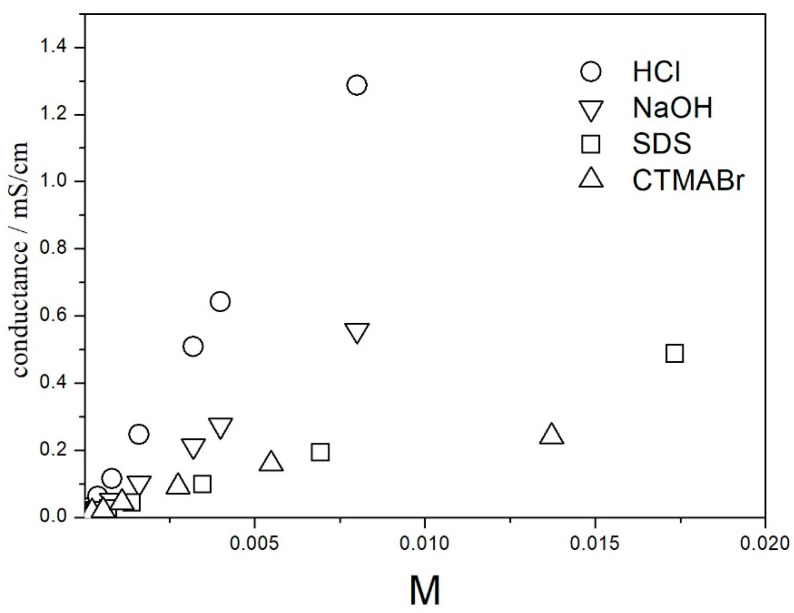
The effect of electrolytes on the conductance of titania dispersion in 50% EG.

**Figure 2 molecules-27-02162-f002:**
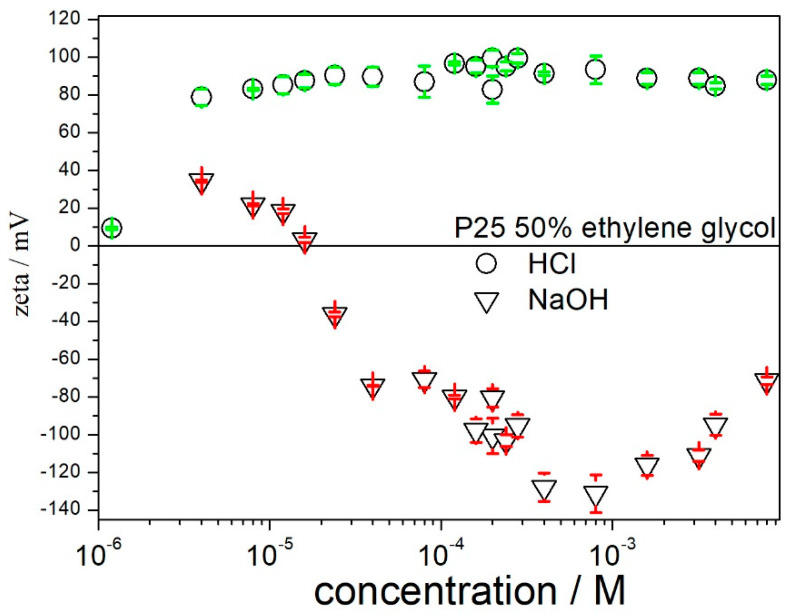
The effect of acid and base on the zeta potential of titania in 50% EG.

**Figure 3 molecules-27-02162-f003:**
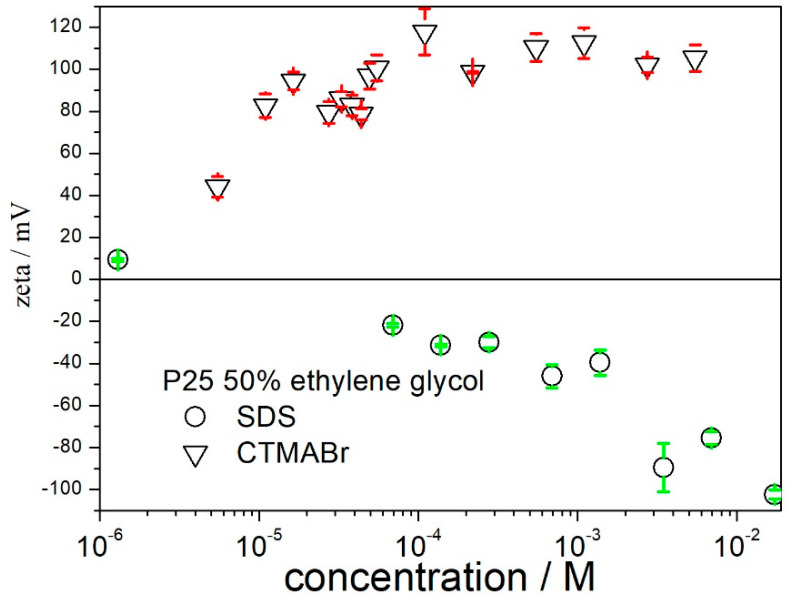
The effect of ionic surfactants on the zeta potential of titania in 50% EG.

**Figure 4 molecules-27-02162-f004:**
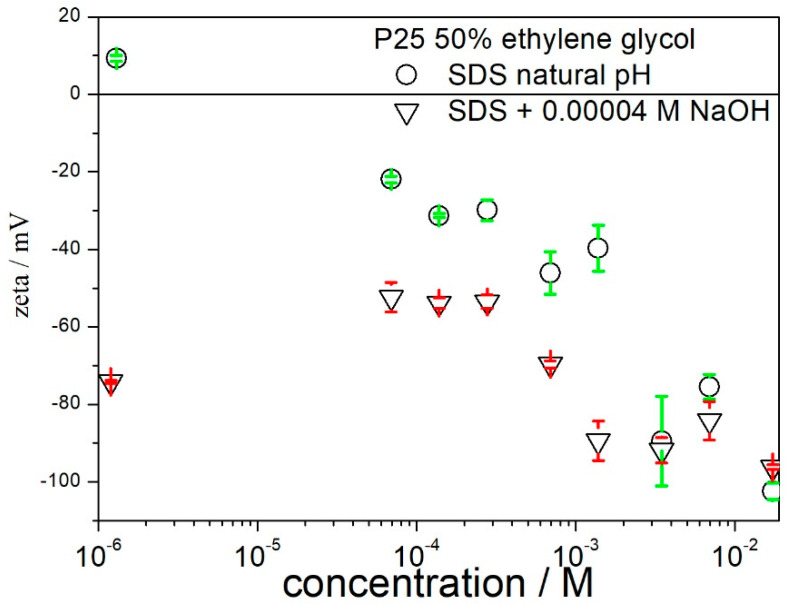
The effect of acidity on the zeta potential of titania in 50% EG in the presence of SDS.

**Table 1 molecules-27-02162-t001:** Studies of zeta potentials in EG–water mixtures.

EG Concentration	Particles	Surfactants and Other Additives	Variables	Ref.
100; 90; 80	MWCNT original and oxidized	CTAB		[6]
100; 90; 80	MWCNT original and oxidized	none		[7]
60; 40; 20	Al_2_O_3_	none		[8]
50	Fe_3_O_4_	SDS, CTAB, SDBS	A	[9]
50	functionalized graphene	PVP		[10]
50	Al_2_O_3_ + SiO_2_	none		[11]
50	MgO	CTAB	SL, A	[12]
50	ZnO	none		[13]
40	TiO_2_, Al_2_O_3_	none	SL, A	[14]
40	TiO_2_, functionalized graphene	SDS, SDBS, CTAB, PVP, SDC, TX-100	A	[15,16]
30	functionalized carbon dots	none	A	[17]
commercial coolant, >50% EG	SiO_2_	none	SL, A	[18]

Abbreviations in Table 1: MWCNT multiwall carbon nanotubes; SDS sodium dodecyl sulfate; CTAB cetyltrimethylammonium bromide; SDBS sodium dodecylbenzenesulfonate; TX-100 Triton X-100, commercial surfactant; PVP polyvinylpyrrolidone; SDC sodium deoxycholate; SL solid load; A aging.

**Table 2 molecules-27-02162-t002:** The κa and f in typical dispersions used in this study. Both κa and f are dimensionless.

*c*/M	*κa*	*f*
0.0002	1.14	1.04
0.0004	1.62	1.05
0.0008	2.29	1.07
0.0016	3.23	1.1
0.0032	4.57	1.14
0.004	5.11	1.15
0.008	7.23	1.21

## Data Availability

Data are available from the corresponding author on request.

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
