# Peer review of "Electric Double Layer in Water-Organic Mixed Solvents: Titania in 50% Ethylene Glycol"

_molecules, 2022, doi:10.3390/molecules27072162_

Round 1

Reviewer 1 Report

The manuscript describes the factor that influences zeta potential (ζ) which is not discovered in the previous research. The author measured the electrophoretic mobility at various surfactant concentrations and the pH-dependent surface charging which are related to zeta potential in a mixture of ethylene glycol (EG) and water. However, more information and experiments must be supplemented in the manuscript to prove the author's arguments mentioned in the introduction part. Therefore, I recommend this manuscript be published in MDPI after a major revision.

Comment 1

In line 160, the author mentioned the ζ potential trend reverses with NaOH concentration of 0.001 M. But even those with the opposing phenomenon, it is stated that both the increase and decrease of ζ potential are caused by the same reason, the addition of base. The author would be better to give a proper reason for each opposing phenomenon.

Comment 2

In line 167, the author insisted that charge regulation by the addition of acid or base is cost-efficient and environment-friendly as compared with the addition of surfactants. That argument needs to be proved by references or quantitative data.

Comment 3

In line 174, when the CTMABr concentration exceeds 0.00002 M, the effect on the ζ potential is insignificant. In contrast, SDS shows a gradual increase in ζ potential. I wonder why SDS and CTMABr show different tendencies by concentration changes.

Comment 4

In the introduction part, the author refuted the claims of previous studies or mentioned cases of recent studies. However, there are no references to them. The author should help readers understand by adding accurate references to related studies. Please add several references for the readers.

Comment 5

Currently, the abstract is insufficient to explain this research. Several main aspects are not revealed in the abstract section, such as the difference between the adding acid/base and the adding surfactants as mentioned throughout the manuscript. I hope the author will clearly describe the subject and results of the study.

Comment 6

In conclusion, it would be better to emphasize what the author intends to argue through this research as mentioned in the introduction. Furthermore, it seems to be a problem with generalization through very few variable adjustments. I hope that the author's argument can be generalized under other conditions through additional experiments and analysis.

Author Response

The manuscript describes the factor that influences zeta potential (ζ) which is not discovered in the previous research. The author measured the electrophoretic mobility at various surfactant concentrations and the pH-dependent surface charging which are related to zeta potential in a mixture of ethylene glycol (EG) and water. However, more information and experiments must be supplemented in the manuscript to prove the author's arguments mentioned in the introduction part. Therefore, I recommend this manuscript be published in MDPI after a major revision.

Comment 1

In line 160, the author mentioned the ζ potential trend reverses with NaOH concentration of 0.001 M. But even those with the opposing phenomenon, it is stated that both the increase and decrease of ζ potential are caused by the same reason, the addition of base. The author would be better to give a proper reason for each opposing phenomenon.

The following explanation was added:

 Addition of base leads to an increase in the ionic strength. This is well-known (cf. ref. 3 and references therein) that the high ionic strength depresses the absolute value of ζ potential. Moreover monovalent cations are specifically adsorbed on titania from water-alcohol mixtures (cf. ref. 3 and references therein), and this can very well be that similar specific adsorption occurs in EG-water mixtures. Specific adsorption of cations leads to more positive (less negative) electrokinetic charge. Thus we have two opposite trends on addition of base: increase in the pH leads to more negative ζ potential, but increase in the ionic strength and specific adsorption of cations leads to less negative ζ potential. Apparently the pH effect prevails at low base concentrations, but the ionic strength effect and/or specific adsorption of cations prevail at high base concentrations.

Comment 2

In line 167, the author insisted that charge regulation by the addition of acid or base is cost-efficient and environment-friendly as compared with the addition of surfactants. That argument needs to be proved by references or quantitative data.

The following explanation was added:

 For example (Sigma-Aldrich online catalogue, accessed March 21, 2022) 1 mole of NaOH (reagent grade, 1 kg jar) costs 1.6 euro while 1 mole of SDS (reagent grade, 1 kg jar) costs 28 euro, and 1 mole of CTMABr (reagent grade, 1 kg jar) costs 106 euro. Many papers have been published on environmental risks related to surfactants [22, and references therein].

Comment 3

In line 174, when the CTMABr concentration exceeds 0.00002 M, the effect on the ζ potential is insignificant. In contrast, SDS shows a gradual increase in ζ potential. I wonder why SDS and CTMABr show different tendencies by concentration changes.

 We added the following. This is only a speculation, and more research is required to make such claims.

 The substantial difference between SDS (HLB of 40) and CTMABr (HLB 10) may be due to the difference in their hydrophylic-lipophylic characters.

Comment 4

In the introduction part, the author refuted the claims of previous studies or mentioned cases of recent studies. However, there are no references to them. The author should help readers understand by adding accurate references to related studies. Please add several references for the readers.

We have published a paper recently (ref. 5) on similar topic, and we would like to avoid repetitions. To make this even more clear we added:

In our opinion the following problems have not received sufficient attention in the previous studies [6-18]

  • The calculation of zeta potential from electrophoretic mobility is not straightforward.
  • The relationship between zeta potential and stability is complicated.
  • More attention should be paid to proper selection of the surfactant and of the surfactant dose.

We address these problems in our study

Comment 5

Currently, the abstract is insufficient to explain this research. Several main aspects are not revealed in the abstract section, such as the difference between the adding acid/base and the adding surfactants as mentioned throughout the manuscript. I hope the author will clearly describe the subject and results of the study.

 The abstract was extended. Please note that in the new instructions for authors the abstract should give a wide background and present experimental methods, so not much space is left for results and discussion.

 We added the following.

Overdose of base (>1 mM) results in depression of the negative zeta potential. This result may be due to specific adsorption of sodium on titania from 50% EG.

Comment 6

In conclusion, it would be better to emphasize what the author intends to argue through this research as mentioned in the introduction. Furthermore, it seems to be a problem with generalization through very few variable adjustments. I hope that the author's argument can be generalized under other conditions through additional experiments and analysis.

We tried to avoid repetitions. Regarding the generalizations: we clearly stated what is the result and what is speculation. We hope that our conclusion section will encourage the others to confirm or challenge our speculations.

Reviewer 2 Report

This article reports an interesting study on electric double layer in water - EG mixed solvents. The manuscript is well written and the results presented are relevant in electrochemistry and would be useful in the field of heat transfer fluids. This manuscript could be accepted for publication in molecules after minor revision.

  1. The introduction is bit long and lacks clarity, it would be better to restructure the introduction part that the readers can easily understand the relevance of this research work. Might be easier if it follows an order such as context of the work, what has been done already, what is missing and what you are trying to study here.
  2. References and citations are missing in many places in the introduction part. Whenever previous work/literature is mentioned in the manuscript, it should have proper citations. Noticed Lines 42-43, 45-46, 48, 71-72, 77-80, 84-85, 86, 93-94 etc.
  3. The data on dielectric constant of aq.EG are very limited as written in materials and methods. Is it very difficult to experimentally determine the dielectric constant of your system so that you can compare with the value shown in literature (90 years old)?

Author Response

This article reports an interesting study on electric double layer in water - EG mixed solvents. The manuscript is well written and the results presented are relevant in electrochemistry and would be useful in the field of heat transfer fluids. This manuscript could be accepted for publication in molecules after minor revision.

  1. The introduction is bit long and lacks clarity, it would be better to restructure the introduction part that the readers can easily understand the relevance of this research work. Might be easier if it follows an order such as context of the work, what has been done already, what is missing and what you are trying to study here.

We have a problem then, because in the editorial review we were asked to extend the Intro. The original Intro was shorter. Regarding clarity. We have published a paper recently (ref. 5) on similar topic, and we would like to avoid repetitions. To make this even more clear we added:

In our opinion the following problems have not received sufficient attention in the previous studies [6-18]

  • The calculation of zeta potential from electrophoretic mobility is not straightforward.
  • The relationship between zeta potential and stability is complicated.
  • More attention should be paid to proper selection of the surfactant and of the surfactant dose.

We address these problems in our study

  1. References and citations are missing in many places in the introduction part. Whenever previous work/literature is mentioned in the manuscript, it should have proper citations. Noticed Lines 42-43, 45-46, 48, 71-72, 77-80, 84-85, 86, 93-94 etc.

done

  1. The data on dielectric constant of aq.EG are very limited as written in materials and methods. Is it very difficult to experimentally determine the dielectric constant of your system so that you can compare with the value shown in literature (90 years old)?

We cannot because they report a value for 20 oC (our study was performed at 25 oC). We explain this in the new Ms.

The later value was based on the original data for EG-water mixtures [27] obtained at 20 oC, and the temperature coefficients taken from [20,26].

Round 2

Reviewer 1 Report

Thank you for your comment and supplementation. Your answer seems reasonable. This manuscript can be published in this form.